# Grade repetition and bullying victimization in adolescents: A global cross-sectional study of the Program for International Student Assessment (PISA) data from 2018

Qiguo Lian[1], Chunyan Yu[1], Xiaowen Tu[1], Minglin Deng[2], Tongjie Wang[2], Qiru Su[3]*, Xiayun Zuo[1]*

1 NHC Key Laboratory of Reproduction Regulation, Shanghai Institute for Biomedical and Pharmaceutical Technologies, Fudan University, Shanghai, China, 2 Jiangyou People's Hospital, Sichuan, China, 3 Shenzhen Children's Hospital, Guangdong, China

* suqiru_sz@163.com (QS); zuoxiayun@sippr.org.cn (XZ)

## Abstract

### Background

Grade repetition is practiced worldwide and varies considerably across the globe. Globally, around 32.2 million students repeated a grade at the primary education level in 2010. Although a large body of research has documented grade repetition's academic and non-academic effects, the limited evidence on associations between grade repetition and school bullying is inconsistent and ambiguous. This study aimed to investigate the global association of grade repetition with bullying victimization in a large-scale school-based cross-sectional study.

### Methods and findings

We used the latest global data from the Program for International Student Assessment (PISA) 2018. PISA 2018 was conducted between March and August 2018 in 80 countries and economies among students aged 15–16 years attending secondary education. The students reported their experiences of repeating a grade at any time point before the survey and of being bullied in the past 12 months. The outcome measures were 6 types of bullying victimization. We accounted for the complex survey design and used multivariate logistic regression models to estimate the odds ratios (ORs) with 95% confidence intervals (CIs) of grade repetition with bullying victimization after adjusting for potential confounders (sex; age group; migrant status; school type; economic, social, and cultural status; and parental emotional support). This study included 465,146 students (234,218 girls and 230,928 boys) with complete data on grade repetition and bullying victimization in 74 countries and economies. The lifetime prevalence of grade repetition was 12.26%, and 30.32% of students experienced bullying at least a few times a month during the past 12 months. Grade repetition was statistically significantly associated with each type of bullying victimization. The OR (95% CI) of overall bullying victimization for grade repeaters compared with their promoted peers

**Data Availability Statement:** All data and materials used in this study are available for download at https://www.oecd.org/pisa/data/2018database/

**Funding:** The author(s) received no specific funding for this work.

**Competing interests:** The authors have declared that no competing interests exist.

**Abbreviations:** CI, confidence interval; ESCS, economic, social, and cultural status; ISCED, International Standard Classification of Education; OECD, Organisation for Economic Co-operation and Development; OR, odds ratio; PISA, Program for International Student Assessment; SDG, Sustainable Development Goal.

was 1.42 (95% CI 1.32–1.52, $p$ < 0.001). The sex-specific analysis produced similar results in both boys and girls. Furthermore, girls who repeated a grade had higher risks of being made fun of, being threatened, having possessions taken away, and being pushed around than boys. The major limitation is that this study only included students attending schools and therefore may be subject to possible selection bias. In addition, the cross-sectional design hinders us from establishing causality between grade repetition and bullying victimization.

## Conclusions

In this study, we observed that, globally, both boys and girls who repeat a grade are at increased risk of being bullied compared with promoted peers, but girls may experience higher risks than boys of specific types of bullying associated with repeating a grade. These findings provide evidence for the association of grade repetition with bullying victimization. Sex differences in risk of experiencing some types of bullying suggest that tailored interventions for girls who repeat a grade may be warranted.

---

## Author summary

### Why was this study done?

- Preventing school violence is a specific target of the United Nations Sustainable Development Goals.

- Few empirical studies involving large samples have examined the association of grade repetition with bullying victimization.

### What did the researchers do and find?

- We examined the association of grade repetition with bullying victimization among 465,146 15-year-old students in 74 countries and economies.

- The prevalence of grade repetition and bullying victimization varied widely across countries and economies.

- Grade repetition was associated with bullying victimization in both boys and girls.

- Girls who repeated a grade experienced higher risks than boys of being made fun of, being threatened, having possessions taken away, and being pushed around.

### What do these findings mean?

- The experience of repeating a grade may suggest a need for bullying interventions among both boys and girls.

- Tailored interventions for girls who repeat a grade may be warranted.

## Introduction

Grade repetition (or grade retention), the practice of having students repeat a grade for an additional school year when they fail to meet required benchmarks, is relatively common in some countries (e.g., Belgium) and is discouraged or banned outright in others (e.g., Norway) [1]. According to data collected by the Program for International Student Assessment (PISA) in 2009, over 10% of 15-year-olds in 30 countries and economies had repeated a grade at least once [2]. Globally, 32.2 million students repeated a grade at the primary education level in 2010 [3]. Grade repetition and its opposite (social promotion) are controversial issues in many countries. The former is viewed as costly [3], with unclear benefits and well-known drawbacks [4–8], and a common misperception is that repeating a grade allows the student to grow academically and socially [9].

Moreover, the practice of grade repetition can have various negative consequences. A US study found that grade repetition was independently associated with increased risk of behavior problems among white children and adolescents [10]. Despite a growing evidence base that links grade repetition to adverse behavioral outcomes [11–13], the literature includes mostly small observational studies of specific behavioral outcomes, such as bullying victimization, and provides inconsistent results [14,15]. One study found no differences in bullying victimization between socially promoted and retained students in public schools [14]. Another found that students who were old for their grade, because of grade retention or delayed school entry, were more likely to be bullied physically, verbally, and socially than their peers who were age appropriate for their grade [15].

Preventing school violence in all its forms is a fundamental human rights issue. Bullying victimization—individuals being repeatedly exposed to intentional harmful behaviors by peers and unable to defend themselves because of a power imbalance [16]—is a severe global public health problem that compromises learners' right to education. Approximately 130 million 13- to 15-year-olds worldwide experience bullying [17]. The prevalence of bullying victimization varies widely among countries; however, one study of adolescents in 83 countries estimated an overall prevalence of 30.5% in the past 30 days [18]. School bullying has short- and long-term adverse consequences, including physical, cognitive, and mental health issues for the victimized students [19] and longstanding economic impacts for the victimized students, their families, and society [20].

The social–ecological framework is useful for understanding the factors that contribute to grade repetition and related bullying victimization [21]. This framework considers the complex interplay between individual, family, school, and societal factors. One potential pathway that links grade repetition to bullying victimization is poor academic performance [14,16]. Academic problems are the major reason for students to repeat a grade [22] and increase the risk of bullying because struggling with academic work could be viewed as deviant behavior by promoted students [14]. The social stigma attached to repeating a grade may also play an important role in bullying in retained subgroups [23,24], and the degree of stigma varies significantly across cultures.

Addressing school bullying is a global policy priority and essential to achieving the Sustainable Development Goals (SDGs), in particular SDG 4.a.2, to provide safe, non-violent, inclusive, and effective learning environments for all, and SDG 16.2, to end all forms of violence against children, including bullying [25]. To date, the association of grade repetition and bullying victimization has not been closely examined in global samples of adolescents, although many studies have estimated the prevalence of grade repetition and bullying victimization separately [2,18]. This study aimed to address the gap by providing a comprehensive overview of the global prevalence of grade repetition and bullying victimization and examining the

association of grade repetition with bullying victimization using the latest PISA data. We hypothesized that grade repetition would be associated with bullying victimization in both boys and girls, and that the association would be stronger in girls because grade retention is socially more stigmatizing for girls [26].

## Methods

### Study design and participants

We used the data from the 2018 survey cycle of PISA. The assessment is conducted by the Organisation for Economic Co-operation and Development (OECD) every 3 years to assess the skills and knowledge of 15-year-old students around the world in reading, mathematics, and science. The PISA target population is all students attending educational institutions in grade 7 or higher, aged 15 years 3 months to 16 years 2 months, with a 1-month variation of this age window at the beginning of each assessment period. Given that 15-year-olds make up the majority of the population, we follow the convention of using "15-year-old students" as shorthand for the PISA target population.

PISA 2018 was conducted between 1 March 2018 and 31 August 2018 in 80 countries and economies worldwide. A 2-stage sampling procedure was adopted within most countries/economies. In the first stage, schools were sampled systematically with probabilities proportional to the estimated size of their 15-year-old student population from a comprehensive national list of all PISA-eligible schools. At least 150 schools were selected within each country/economy. In the second stage, 42 15-year-old students were selected with equal probability from each school. If fewer than 42 students were available at a school, all 15-year-old students in the school were sampled. The number of sampled students could not fall below 20 [27]. PISA's international protocol requires a minimum weighted response rate of 85% of sampled schools within participating countries/economies and 80% of students within selected schools. A detailed description of sampling procedures, data collection methodology, quality assurance, and data access is available on the official website [28]. Each participating student had 2 hours to complete the tests and about 35 minutes to answer a background questionnaire. In total, 612,002 students completed the PISA 2018 assessments, representing about 32 million 15-year-old students in the participating 80 countries/economies [27].

We used the PISA 2018 dataset in the present study because it is the most recent publicly accessible PISA dataset that measured both grade repetition and bullying victimization. We merged the PISA 2018 school dataset and student dataset and included all students ($n$ = 612,002) who participated in PISA 2018, as shown in Fig 1. All participants in Israel, Lebanon, and North Macedonia were excluded because the 3 countries did not measure school bullying ($n$ = 17,806). All participants in Japan, Malaysia, and Norway were excluded because these 3 countries do not allow grade repetition, and no students reported being retained ($n$ = 18,033). Lastly, the participants without data on grade repetition or school bullying ($n$ = 111,017) were excluded from the analysis. The final sample consisted of 465,146 students, including 234,218 girls and 230,928 boys, in 74 countries/economies worldwide. The list of participating countries/economies and the corresponding sample sizes are provided in S1 Table.

The present study followed the reporting guidelines of Strengthening the Reporting of Observational Studies in Epidemiology (STROBE) for cross-sectional studies (S2 Table). This study was exempt from institutional review board approval because the data used in this secondary analysis are de-identified and publicly available [28].

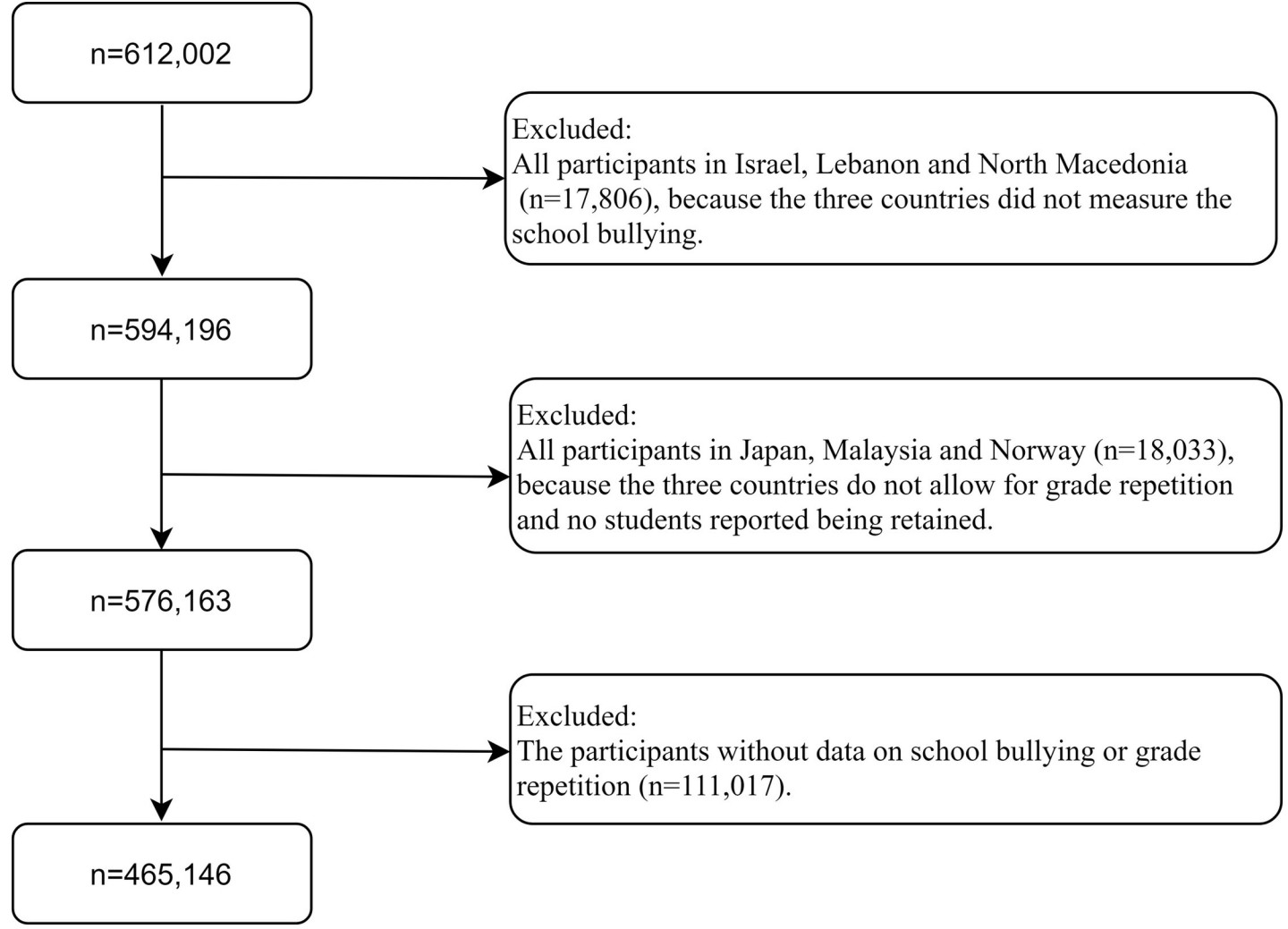

**Fig 1. Participant flow chart.**

## Exposure measurement

Given that education systems vary considerably across countries, PISA followed the widely used International Standard Classification of Education (ISCED) [29] to facilitate comparisons of education systems across participating countries and economies.

Standardized questionnaires asked students 3 questions about their experience with grade repetition: (1) "Have you ever repeated a grade at ISCED 1 (the primary education level)?"; (2) "Have you ever repeated a grade at ISCED 2 (the lower secondary education level)?"; and (3) "Have you ever repeated a grade at ISCED 3 (the upper secondary education level)?" The response options were "No, never", "Yes, once", or "Yes, twice or more" for each of these 3 questions.

Participants were categorized as grade repeaters if they answered "yes" to any of these 3 questions and were categorized as grade non-repeaters (promoted students) if they did not report the experience of repeating a grade at school before the survey.

## Outcome measurement

PISA collected data on students' exposures to school bullying with the question "During the past 12 months, how often have you had the following experiences in school? (Some experiences can also happen in social media)." The listed experiences were as follows: (1) "Other students left me out of things on purpose"; (2) "Other students made fun of me"; (3) "I was threatened by other students"; (4) "Other students took away or destroyed things that belong to me"; (5) "I got hit or pushed around by other students"; and (6) "Other students spread nasty rumors about me" [30]. The response options were "0 = Never or almost never"; "1 = A few times a year"; "2 = A few times a month"; or "3 = Once a week or more" (Cronbach's alpha = 0.88 in our sample).

Following the cutoff point recommended by previous studies [31,32], for each of the 6 individual bullying experiences mentioned above, we combined the response options for "0 = Never or almost never" and "1 = A few times a year" into 0 (a few times a year or less frequently) and combined the response options for "2 = A few times a month" and "3 = Once a week or more" into 1 (a few times a month or more frequently).

We then combined the 6 individual bullying experiences into a single indicator "any type of bullying victimization" and coded it as 0 (never or rare on all types) if the participants reported "0 = Never or almost never" or "1 = A few times a year" to all of the 6 experiences and 1 (any involvement) if they reported "2 = A few times a month" or "3 = Once a week or more" to any of the 6 experiences.

## Covariates

The ecological system framework was used to identify potential confounders of the association between grade repetition or promotion and bullying. The covariates included sex; age group; migrant status; school type; economic, social, and cultural status (ESCS); and parental emotional support.

Students in PISA 2018 were asked the question "Are you female or male?" We generated a dichotomous variable "sex" with 1 = female and 2 = male based on this question.

We calculated student age based on the assessment date and birthday provided by the participating students [33]. As suggested by a previous PISA study [2], we generated a dichotomous variable "age group" with 1 indicating an individual's age is below the average age of the country's PISA sample and 2 indicating an individual's age is at or above this average age.

Students were asked the country of birth of themselves and their parents by the question "In what country were you and your parents born?" We generated a dichotomous variable "migrant status" with 1 = native students and 2 = migrant students. Native students are those students born in the country of assessment with at least 1 parent also born in the same country, while migrant students include those born in the country of assessment but whose parents were born in another country (second-generation migrants) and those born outside the country of assessment and whose parents were also born in another country (first-generation migrants).

The school principals were asked the question "Is your school a public or a private school?" with the responses 1 = a public school and 2 = a private school. Public school means the school is managed directly or indirectly by a public education authority, government agency, or governing board appointed by the government or elected by a public franchise, while private school indicates the school is managed directly or indirectly by a non-government organization (e.g., church, trade union, business, or other private institution).

The ESCS of the students was derived from the variables about their parental highest occupational status, parental educational attainment, and home possessions. The parental

occupational status was obtained from responses to 2 open-ended questions: (1) "What's your mother's/father's main job?" and (2) "What does your mother/father do in her/his main job?" The responses were coded and mapped to the International Socio-Economic Index of Occupational Status (ISEI), with higher ISEI scores indicating higher levels of occupational status. The parental educational status was generated by the questions (1) "What is the <highest level of schooling> completed by your mother/father?" and (2) "Does your mother/father have any of the following qualifications?" The index of home possessions was constructed by self-reported availability of 16 household items at home and amount of possessions and books at home. ESCS was computed by attributing equal weight to the 3 standardized components. The value of ESCS has a mean of 0 and a standard deviation of 1 across OECD countries [2]. More details on how ESCS was constructed can be found in the PISA 2018 Technical Report [33].

The variable "parental emotional support" was derived from responses to the following statements: (1) "My parents support my educational efforts and achievements"; (2) "My parents support me when I am facing difficulties at school"; and (3) "My parents encourage me to be confident." The response options were "strongly disagree," "disagree," "agree," and "strongly agree." The responses to these 3 statements were combined to create the index of perceived emotional support from parents using the item response theory scaling model; the average of the index is 0, and the standard deviation is 1 across OECD countries [30]. The details of the derived index are documented in the PISA 2018 Technical Report [33] and the PISA 2015 Technical Report [34]. The PISA questions used to obtain data on exposure, outcome, and covariates can be found in S3 Table.

## Statistical analysis

There was no documented analytical protocol in PISA-related documents to be followed in addressing our specific research question. An analysis plan was created at the beginning of November 2020, after we identified our research question. The analyses were conducted between 9 November 2020 and 26 July 2021, and no data-driven changes to analyses took place.

We performed all statistical analyses using svy: prefix commands in Stata/SE software (version 15.1; StataCorp), and the tests were considered statistically significant if the 2-sided $p$-value was less than 0.05. We used the *svyset* command to account for the complex survey sampling design and sampling weight (w_fstuwt). The primary sampling unit is generated by country ID and international school ID. We included replication weights (wfstr1–wfstr80) and estimated the standard errors using balanced repeated replication with Fay's adjustment [35,36]. We also specified the *mse* option in the *svyset* command to request the mean squared error version of the estimator [35].

We first calculated the percentage of the students' individual characteristics, then estimated the prevalence of grade repetition and each type of bullying victimization by sex. We then estimated the sex-stratified prevalence of each type of bullying victimization among promoted students and grade repeaters. Then, we used multivariable logistic regression models to estimate the odds ratios (ORs) and corresponding 95% confidence intervals (CIs) of the associations of grade repetition with each type of bullying victimization by sex, adjusting for country differences and the aforementioned control variables. Given the underlying sex differences in grade repetition and bullying victimization, we examined the association of grade repetition and sex interaction with bullying victimization. Considering that the participating students in the study were from diverse countries with different cultures, we further reproduced the analyses of estimation and association at the country level.

To evaluate the direction and magnitude of unmeasured confounding, due to this being an observational study, we used the community-contributed command *evalue* [37] to report the

*E*-value (a new measure for performing sensitivity analyses) after the multivariable logistic regression models, as suggested by VanderWeele and Ding and colleagues [38,39]. The recently proposed *E*-value quantifies the minimum strength of association with both the exposure and the outcome that would be required of an unmeasured confounder to fully explain away the observed effect [38,40].

## Results

This study included 465,146 students from 74 countries/economies, with a mean age of 15.81 years (range 15.08–16.33 years) and a standard deviation of 0.29 years (the average ages by country are listed in S4 Table); 234,218 (50.31%) participating students were girls, and 230,928 (49.69%) were boys (Table 1). Also, 7.57% were migrants and 18.71% attended private schools.

On average across participating countries/economies, the prevalence of grade repetition was 12.26% (95% CI 11.80%–12.72%). Overall, 13.72% (95% CI 13.38%–14.07%) of students reported being frequently (a few times a month or more) left out things, 17.92% (95% CI 17.54%–18.32%) reported being frequently made fun of, 10.05% (95% CI 9.73%–10.37%) reported being frequently threatened, 12.03% (95% CI 11.66%–12.42%) reported having possessions taken away frequently, 10.45% (95% CI 10.08%–10.83%) reported being frequently hit or pushed around, 14.01% (95% CI 13.67%–14.37%) reported being a subject of rumors frequently, and 30.32% (95% CI 29.83%–30.81%) reported frequently experiencing any type of bullying. There were statistically significant differences in the prevalence of grade repetition and each bullying type between boys and girls. Boys were more likely to be retained and bullied compared to girls (Table 2).

As shown in Table 3, grade repetition was associated with each type and any type of bullying victimization before adjustment for measured confounders. Compared with promoted peers, grade repeaters were more likely to experience being frequently left out (OR = 1.46 [95% CI 1.36–1.57]), being frequently made fun of (OR = 1.44 [95% CI 1.35–1.55]), being frequently threatened (OR = 2.26 [95% CI 2.11–2.42]), having possessions taken away frequently (OR = 2.10 [95% CI 1.95–2.25]), being frequently hit or pushed around (OR = 2.27 [95% CI 2.11–2.45]), being a subject of rumors frequently (OR = 2.02 [95% CI 1.91–2.15), and frequently experiencing any type of bullying (OR = 1.68 [95% CI 1.58–1.78]) (all *p*-values < 0.001). The sex-specific analyses showed similar results in boys and girls.

**Table 1. Descriptive statistics of the individual characteristics.**

| Characteristic | Total sample (*n* = 465,146) | Girls (*n* = 234,218) | Boys (*n* = 230,928) |
|---|---|---|---|
| **Age group** | | | |
| At or above country average age | 229,302 (49.67) | 115,300 (49.60) | 114,002 (49.74) |
| Below country average age | 235,844 (50.33) | 118,918 (50.40) | 116,926 (50.26) |
| **Migrant status** | | | |
| Native | 396,370 (92.43) | 200,578 (92.49) | 195,792 (92.38) |
| Migrant | 58,593 (7.57) | 29,380 (7.51) | 29,213 (7.62) |
| **School type** | | | |
| Private | 82,945 (18.71) | 41,606 (18.27) | 41,339 (19.16) |
| Public | 354,351 (81.29) | 178,777 (81.73) | 175,574 (80.84) |
| **Index of economic, social, and cultural status** | −0.61 (1.24) | −0.63 (1.26) | −0.65 (1.27) |
| **Index of parental emotional support** | −0.10 (1.01) | −0.03 (1.00) | 0.02 (0.99) |

Data are *n* (weighted percent) or weighted mean (SD).

**Table 2. Prevalence of grade repetition and bullying victimization among girls and boys.**

| Outcome | Prevalence of outcome, *n* (weighted percent) | | | OR (95% CI, *p*-value) |
|---|---|---|---|---|
| | Total sample (*n* = 465,146) | Girls (*n* = 234,218) | Boys (*n* = 230,928) | |
| **Grade repetition** | 46,773 (12.26) | 19,019 (9.53) | 27,754 (15.01) | 1.68 (1.58–1.78, <0.001) |
| **Type of bullying** | | | | |
| Other students left me out of things on purpose | 52,410 (13.72) | 23,733 (12.61) | 28,677 (14.85) | 1.21 (1.15–1.27, <0.001) |
| Other students made fun of me | 70,499 (17.92) | 28,905 (15.67) | 41,594 (20.22) | 1.36 (1.29–1.44, <0.001) |
| I was threatened by other students | 39,736 (10.05) | 13,304 (7.37) | 26,432 (12.77) | 1.84 (1.71–1.98, <0.001) |
| Other students took away or destroyed things that belonged to me | 44,327 (12.03) | 15,465 (9.37) | 28,862 (14.74) | 1.67 (1.57–1.78, <0.001) |
| I got hit or pushed around by other students | 41,899 (10.45) | 13,175 (7.35) | 28,724 (13.60) | 1.98 (1.85–2.13, <0.001) |
| Other students spread nasty rumors about me | 58,004 (14.01) | 25,201 (12.40) | 32,803 (15.66) | 1.31 (1.24–1.39, <0.001) |
| Any type of the above | 124,103 (30.32) | 53,513 (27.06) | 70,59 (33.61) | 1.36 (1.31–1.43, <0.001) |

OR, odds ratio.

After adjustment for sex, age group, migrant status, school type, ESCS, and parental emotional support, the OR of each type of bullying victimization for grade repeaters compared with promoted students was significantly greater than 1: 1.27 (95% CI 1.17–1.38) for being frequently left out, 1.20 (95% CI 1.11–1.31) for being frequently made fun of, 1.69 (95% CI 1.56–1.83) for being frequently threatened, 1.61 (95% CI 1.47–1.76) for having possessions taken away frequently, 1.70 (95% CI 1.56–1.85) for being frequently hit or pushed around, 1.65 (95% CI 1.53–1.77) for being a subject of rumors frequently, and 1.42 (95% CI 1.32–1.52) for frequently experiencing any type of bullying (all *p*-values < 0.001). The sex-specific analyses produced similar results in boys and girls (Table 4). Furthermore, while sex differences were not apparent overall for bullying victimization, there were interactions indicating that sex differences existed for some types of bullying. Specifically, girls who repeated a grade had higher risk than boys of being made fun of, being threatened, having possessions taken away, and being pushed around (S5 Table).

In the sensitivity analysis, the observed associations between grade repetition and bullying victimization seemed moderately robust to potential unmeasured confounding, according to the calculated *E*-values and corresponding 95% CIs (Table 4). For instance, to explain away an OR of 1.42 for any type of bullying victimization, an unmeasured confounder associated with both grade repetition and bullying victimization of any type with an OR of 2.19 each could suffice, but weaker confounding could not. To shift the CI lower bound to contain the null value, an unmeasured confounder associated with both grade repetition and bullying victimization of any type with an OR of 1.97 could suffice, but weaker confounding could not.

In the country-specific analysis, the prevalence estimates for grade repetition and any type of bullying victimization varied widely across the participating countries/economies (grade repetition range 0.81%–42.53%; bullying victimization range 9.4%–64.81%) (S6 Table). Also, grade repeaters reported a higher prevalence of any type of bullying victimization than promoted peers. These findings were true for all countries/economies (S7 Table). The

**Table 3. Prevalence of bullying victimization among promoted students and grade repeaters.**

| Type of bullying | Prevalence of outcome, as *n* (weighted percent), or OR (95% CI, *p*-value) | | |
|---|---|---|---|
| | Total sample (*n* = 465,146) | Girls (*n* = 234,218) | Boys (*n* = 230,928) |
| **Other students left me out of things on purpose** | | | |
| Promoted students | 45,480 (13.12) | 21,189 (12.16) | 24,291 (14.16) |
| Grade repeaters | 6,930 (18.07) | 2,544 (16.90) | 4,386 (18.83) |
| OR (95% CI, *p*-value) | 1.46 (1.36–1.57, <0.001) | 1.47 (1.32–1.64, <0.001) | 1.41 (1.30–1.52, <0.001) |
| **Other students made fun of me** | | | |
| Promoted students | 61,437 (17.22) | 25,654 (15.05) | 35,783 (19.56) |
| Grade repeaters | 9,062 (23.11) | 3,251 (21.64) | 5,811 (24.06) |
| OR (95% CI, *p*-value) | 1.44 (1.35–1.55, <0.001) | 1.56 (1.42–1.71, <0.001) | 1.30 (1.20–1.41, <0.001) |
| **I was threatened by other students** | | | |
| Promoted students | 25,654 (15.05) | 11,276 (6.60) | 21,695 (11.48) |
| Grade repeaters | 3,251 (21.64) | 2,028 (14.93) | 4,737 (20.24) |
| OR (95% CI, *p*-value) | 2.26 (2.11–2.42, <0.001) | 2.48 (2.16–2.86, <0.001) | 1.96 (1.80–2.13, <0.001) |
| **Other students took away or destroyed things that belonged to me** | | | |
| Promoted students | 36,977 (10.89) | 13,228 (8.52) | 23,749 (13.45) |
| Grade repeaters | 7,350 (20.40) | 2,237 (17.67) | 5,113 (22.18) |
| OR (95% CI, *p*-value) | 2.10 (1.95–2.25, <0.001) | 2.31 (2.05–2.60, <0.001) | 1.83 (1.68–2.00, <0.001) |
| **I got hit or pushed around by other students** | | | |
| Promoted students | 34,922 (9.30) | 11,210 (6.61) | 23,712 (12.21) |
| Grade repeaters | 6,977 (18.90) | 1,965 (14.58) | 5,012 (21.69) |
| OR (95% CI, *p*-value) | 2.27 (2.11–2.45, <0.001) | 2.41 (2.11–2.76, <0.001) | 1.99 (1.82–2.18, <0.001) |
| **Other students spread nasty rumors about me** | | | |
| Promoted students | 48,911 (12.80) | 21,936 (11.58) | 26,975 (14.12) |
| Grade repeaters | 9,093 (22.90) | 3,265 (20.29) | 5,828 (24.58) |
| OR (95% CI, *p*-value) | 2.02 (1.91–2.15, <0.001) | 1.94 (1.76–2.15, <0.001) | 1.98 (1.84–2.14, <0.001) |
| **Any type of the above** | | | |
| Promoted students | 107,383 (28.88) | 47,442 (25.99) | 59,941 (32.00) |
| Grade repeaters | 16,720 (40.56) | 6,071 (37.21) | 10,649 (42.72) |
| OR (95% CI, *p*-value) | 1.68 (1.58–1.78, <0.001) | 1.69 (1.55–1.84, <0.001) | 1.58 (1.48–1.69, <0.001) |

OR, odds ratio.

multivariable logistic regression showed that in 46 countries/economies (62.16%), grade repetition was statistically associated with any type of bullying victimization (S8 Table).

## Discussion

To our knowledge, this global analysis of bullying victimization among adolescents is the first to examine the association of grade repetition with bullying using data that were collected with a common international protocol. Our study contributes 3 key findings to the literature. First, there was a high prevalence of grade repetition and bullying victimization observed among the 465,146 participating students, and prevalence varied widely across the 74 countries/economies. Second, grade repeaters were significantly more likely to experience bullying victimization than promoted students. This association was highly consistent between boys and girls. Third, girls who repeated a grade were more likely to report having experienced specific types of bullying than boys.

The prevalence of bullying victimization observed in PISA was consistent with previous studies. A study conducted between 2009 and 2015 reported that the prevalence of being

**Table 4. Associations of grade repetition with bullying victimization for girls and boys.**

| Type of bullying victimization | Adjusted OR (95% CI, *p*-value) or *E*-value (95% CI lower bound) | | |
|---|---|---|---|
| | **Total sample** | **Girls** | **Boys** |
| **Other students left me out of things on purpose** | | | |
| Adjusted OR (95% CI, *p*-value) | 1.27 (1.17–1.38, <0.001) | 1.28 (1.13–1.45, <0.001) | 1.26 (1.15–1.40, <0.001) |
| *E*-value (95% CI lower bound) | 1.86 (1.62) | 1.88 (1.51) | 1.83 (1.57) |
| **Other students made fun of me** | | | |
| Adjusted OR (95% CI, *p*-value) | 1.20 (1.11–1.31, <0.001) | 1.32 (1.18–1.47, <0.001) | 1.15 (1.05–1.27, <0.001) |
| *E*-value (95% CI lower bound) | 1.69 (1.46) | 1.97 (1.64) | 1.57 (1.28) |
| **I was threatened by other students** | | | |
| Adjusted OR (95% CI, *p*-value) | 1.69 (1.56–1.83, <0.001) | 1.93 (1.63–2.30, <0.001) | 1.59 (1.44–1.75, <0.001) |
| *E*-value (95% CI lower bound) | 2.77 (2.50) | 3.27 (2.64) | 2.56 (2.24) |
| **Other students took away or destroyed things that belonged to me** | | | |
| Adjusted OR (95% CI, *p*-value) | 1.61 (1.47–1.76, <0.001) | 1.93 (1.67–2.23, <0.001) | 1.48 (1.33–1.65, <0.001) |
| *E*-value (95% CI lower bound) | 2.60 (2.30) | 3.27 (2.73) | 2.32 (1.99) |
| **I got hit or pushed around by other students** | | | |
| Adjusted OR (95% CI, *p*-value) | 1.70 (1.56–1.85, <0.001) | 1.91 (1.63–2.24, <0.001) | 1.62 (1.47–1.80, <0.001) |
| *E*-value (95% CI lower bound) | 2.79 (2.50) | 3.23 (2.64) | 2.62 (2.30) |
| **Other students spread nasty rumors about me** | | | |
| Adjusted OR (95% CI, *p*-value) | 1.65 (1.53–1.77, <0.001) | 1.63 (1.45–1.83, <0.001) | 1.64 (1.49–1.79, <0.001) |
| *E*-value (95% CI lower bound) | 2.69 (2.43) | 2.64 (2.26) | 2.66 (2.34) |
| **Any type of the above** | | | |
| Adjusted OR (95% CI, *p*-value) | 1.42 (1.32–1.52, <0.001) | 1.49 (1.33–1.66, <0.001) | 1.37 (1.26–1.48, <0.001) |
| *E*-value (95% CI lower bound) | 2.19 (1.97) | 2.34 (1.99) | 2.08 (1.83) |

All models adjusted for country; sex; age group; migrant status; school type; economic, social, and cultural status; and parental emotional support. OR, odds ratio.

bullied at least once in the past 30 days was 34.4% in 68 low- and middle-income countries [41]. Another similar study conducted between 2003 and 2015 in 83 low-, middle-, and high-income countries reported that the prevalence of bullying victimization on 1 or more days in the past 30 days was 30.5% [18]. However, these 2 studies used a generic measure of school bullying. Using multiple specific items with similar cutoffs [42,43], our reported prevalence of any type of frequent bullying of 30.32% is much higher than the value reported by a large study conducted in 2005–2006 in 40 European and North American countries, which revealed a prevalence of bullying of 16.2% in the past 2 months [44]. However, consistent with previous studies [18,44], there was wide variation in the estimated prevalence of bullying victimization between countries.

According to the 2012 Global Education Digest of the United Nations Educational, Scientific and Cultural Organization [3], 32.2 million pupils in 2010 repeated a grade in ISCED 1 and 14.1 million repeated a grade in ISCED 2 globally. An earlier study based on PISA 2009 reported that about 13% of students in OECD countries repeated at least 1 grade, although the prevalence by country ranged from 0% to over 40% [2]. Our reported overall prevalence of 12.26% is therefore consistent with previous studies [2], and we also found wide variation in prevalence across countries.

Research into the consequences of grade repetition has so far focused on its academic outcomes, including academic achievement [2], educational attainment [45], and achievement motivation [5]. However, its non-academic consequences on adolescents' self-esteem [46–48], peer relationships [46], and psychological adjustment [15] have received less attention. Studies

on the association between grade repetition and bullying victimization are especially scarce, and so far have produced inconsistent findings. One study of 378 students in grades 5 to 9 in southern Brazil observed no differences in bullying victimization, including self-reported verbal, physical, and social bullying, between grade repeaters and promoted students [14]. The authors attributed the null findings to the relatively greater physical size of grade repeaters and the normality of being retained in Brazilian public schools [14]. However, a US study with 276 students found that students who were old for their grade were rated by teachers as more likely to be bullied physically, verbally, or socially than their peers who were age appropriate for their grade [15]. The authors speculated that needing to be retained might be considered a red flag for behavioral and learning disabilities, although the intervention of repeating a grade has the negative effect on their social and emotional adjustments [15]. The US study was limited by a lack of distinguishing grade repeaters from their delayed-entry peers [14]. Our study therefore adds to a sparse body of evidence that links grade repetition to bullying victimization. Globally, grade repetition is associated with bullying victimization. Consistent with our hypothesis, we observed that grade repeaters, either boys or girls, were more likely to report being bullied for each type of bullying and any type of bullying.

Although the pathways that underly this association remain unclear, given the study design and coverage of PISA questionnaires, according to the social–ecological framework we posit that an increased risk of victimization is partly attributed to social marginalization and perceived power imbalance between grade repeaters and promoted peers. The stigma associated with "failing" a grade may be a sign of disgrace that alienates the student from the dominant "in group" [24]. Because being retained is often perceived as an academic failure in the school community or society, grade repeaters could be labeled as deviant and stigmatized as failing [49] and could therefore be more vulnerable to bullying than promoted peers [23]. It is important to differentiate the experience of verbal and social bullying from that of physical violence because students who have been retained are more likely to be bullied verbally or socially due to their relatively greater physical size [14]. The counterintuitive finding on physical victimization in our study could be explained by the perceived power imbalance of grade repeaters. According to Olweus, the perceived imbalance is associated with not only physical attributes (e.g., strength, size, and numbers) but also social status (e.g., popularity and preference) in the peer group [16]. However, the increased risks of bullying victimization among grade repeaters could also be partly attributed to the fact they have more difficulty adjusting to their new environment than promoted students who may have lower academic achievement [50]. Also, the significant negative coefficient found for the interaction of grade repetition and sex indicates that girls have a higher risk than boys of some types of bullying victimization associated with repeating a grade. The higher risk of experiencing some types of bullying among girls who repeat a grade suggests the possibility that targeted anti-bullying prevention interventions may be especially beneficial for girls.

The country-specific analyses showed some inconsistency in the association of grade repetition with any type of bullying victimization, although the association was statistically significant in most countries. There are a few potential reasons for these country differences that correspond with the societal level of the social–ecological framework. First, it is possible that the prevalence of bullying victimization is lower where there are bullying prevention policies and programs [44]. Second, cultural differences in what constitutes bullying and tacit social controls over the behavior could also contribute to the variation [51]. Third, the cultural acceptability of violence, violent crime rates, and the degree of stigma for repeating a grade at the country level would also contribute [24,52]. Unfortunately, we did not have data on these indicators to examine these social and cultural mechanisms.

## Strengths and limitations

The main strength of this study is the large representative samples of 74 countries/economies, which enabled us to thoroughly analyze the sex-stratified association of grade repetition with each type of bullying victimization at the global level. In addition, the PISA methodology represents a collaborative standardized procedure on sampling, questionnaire design, and data collection, which generates comparative results across participating countries and economies.

There are also potential limitations that should be considered. First, this study is susceptible to possible selection bias because only adolescents attending schools participate in PISA. Students who are absent or drop out of school are more likely to be bullied and retained [6,19]. Second, the cross-sectional design used in this study hinders us from establishing causality. Although grade repetition may increase the risk of being bullied, bullying victimization could increase the likelihood of repeating a grade because of reduced academic performance [53,54]. Hence our results should be viewed as exploratory and in need of replication using longitudinal data. Third, our data were derived solely from PISA questionnaires, and bias due to unmeasured confounders, including body mass index, disability, ethnicity, and school location, cannot be ruled out. To address this limitation, we calculated *E*-values to quantify the potential unmeasured confounding necessary to nullify our findings [38]. The sensitivity analyses with the *E*-values suggest that our findings are robust against unmeasured confounding. Fourth, the PISA measurement is retrospective self-report and therefore subject to recall bias, although self-report is an accepted method of measuring experiences of bullying and grade repetition [2,16]. Fifth, this study only included 15-year-old adolescents in school, making our findings difficult to generalize to adolescents of other age groups [41].

## Implications

To achieve the global SDG targets for children and adolescents, urgent action is needed to tackle grade repetition and school bullying [25]. If generalizable to broader age groups, our findings may have significant implications for the well-being of 1.3 billion school-age children and adolescents worldwide in an era that increasingly embraces evidence-based interventions [55]. Our study provides evidence for the association of grade repetition with bullying victimization. The observed link raises the possibility that the widespread educational policy of grade repetition may partly contribute to differences in bullying victimization and adds to the evidence against the policy of grade repetition; however, our study cannot establish a causal relationship. Also, sex differences may require particular interventions for girls, although all students who repeat a grade need attention. These results are of great concern for parents, teachers, principals, and policymakers at different levels, especially in countries where grade repetition is particularly prevalent.

## Conclusions

This global analysis of school-age children found that grade repetition and bullying victimization are prevalent, vary widely across countries, and are significantly related. Grade repetition is associated with increased likelihood of bullying victimization, and girls are more likely than boys to experience specific types of bullying associated with repeating a grade.

## Supporting information

**S1 Table. Sample size in this study, by country/economy.**
(DOCX)

**S2 Table. STROBE checklist for cross-sectional studies.**
(DOCX)

**S3 Table. Measure-related questions in PISA 2018.**
(DOCX)

**S4 Table. Country-specific description of age.**
(DOCX)

**S5 Table. Associations of grade repetition and sex interaction with bullying victimization.**
(DOCX)

**S6 Table. Country-specific prevalence of grade repetition and any type of bullying victimization (weighted percent).**
(DOCX)

**S7 Table. Country-specific prevalence of any type of bullying victimization among promoted students and grade repeaters (weighted percent).**
(DOCX)

**S8 Table. Country-specific association between grade repetition and any type of bullying victimization.**
(DOCX)

## Acknowledgments

We thank the OECD for making the PISA data publicly available for analysis. We also thank Prof. Frank J. Elgar, PhD (McGill University, Montreal, Canada), for editing assistance. Prof. Elgar was not compensated for his work.

## Author Contributions

**Conceptualization:** Qiru Su, Xiayun Zuo.

**Formal analysis:** Qiguo Lian.

**Resources:** Minglin Deng, Tongjie Wang.

**Software:** Chunyan Yu.

**Supervision:** Qiru Su, Xiayun Zuo.

**Validation:** Chunyan Yu, Xiayun Zuo.

**Visualization:** Chunyan Yu.

**Writing – original draft:** Qiguo Lian.

**Writing – review & editing:** Chunyan Yu, Xiaowen Tu, Minglin Deng, Tongjie Wang, Qiru Su, Xiayun Zuo.

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
