## [Editor Report · Decision Letter 0]

29 Jan 2021

Dear Dr Zuo, 

Thank you for submitting your manuscript entitled "Grade repetition and bullying victimization in teenagers: a global cross-sectional study" for consideration by PLOS Medicine.

Your manuscript has now been evaluated by the PLOS Medicine editorial staff as well as by the Special Issue guest editors and I am writing to let you know that we would like to send your submission out for external peer review.

Kind regards,

Artur A. Arikainen

Associate Editor

PLOS Medicine

---

## [Decision Letter · Decision Letter 1]

14 May 2021

Dear Dr. Zuo,

Thank you very much for submitting your manuscript "Grade repetition and bullying victimization in teenagers: a global cross-sectional study" (PMEDICINE-D-21-00390R1) for consideration in PLOS Medicine’s Special Issue: Global Child Health: From Birth to Adolescence and Beyond.

Your paper was evaluated by a senior editor and discussed among all the editors here. It was also discussed with the special issue guest editors, and sent to independent reviewers, including a statistical reviewer. The reviews are appended at the bottom of this email and any accompanying reviewer attachments can be seen via the link below:

[LINK]

In light of these reviews, I am afraid that we will not be able to accept the manuscript for publication in the journal in its current form, but we would like to consider a revised version that addresses the reviewers' and editors' comments. Obviously we cannot make any decision about publication until we have seen the revised manuscript and your response, and we plan to seek re-review by one or more of the reviewers. 

We expect to receive your revised manuscript by Jun 04 2021 11:59PM. Please email us (plosmedicine@plos.org) if you have any questions or concerns.

We look forward to receiving your revised manuscript. 

Sincerely,

Caitlin Moyer, Ph.D.

Associate Editor 

PLOS Medicine

plosmedicine.org

1. Abstract: Background: Line 24 (and throughout): Please refer to low or middle income countries rather than "developing countries" and high income countries rather than "developed" countries.

2. Abstract: Methods and findings: Please clarify the age of the students (at line 32 this is indicated as 15 years, but at line 41 a range of 15.08-16.33 years is given). number of participants, years during which the study took place, length of follow up, and main outcome measures.

3. Abstract: Conclusions: Line 57-58: Here, and throughout the manuscript, please avoid statements implying causality of the results. “These findings provide robust evidence for the negative effect of grade repetition on bullying victimization.”

4. Author summary: Please format the author summary with bullet points rather than paragraph form (2-3 points per section). Please revise the final statement to avoid overstating what can be concluded from your study: “Globally the policy of grade repetition is harmful to students’ health and wellbeing in some way regarding to school bullying.”

5. Methods: Line 139-142: Please at least briefly mention how schools and students were selected for inclusion: “A two stage sampling procedure was adopted within each country/economy. In the first stage, a representative sample of at least 150 schools was selected, then roughly 42 students were selected from each school [22]. At line 149-150, please clarify this statement further: “The threshold for an acceptable participation rate after replacement was 85% [22].”

6. Methods: LIne 165-166: make sure that the Methods section transparently describes when analyses were planned, and when/why any data-driven changes to analyses took place (with rationale).

7. Methods: Line 198: Please revise if this should be “Are you female or male?” in this question.

8. Methods: Line 217-219: Please clarify the measures used to derive ESCS.

9. Methods: Line 220-225: Please provide more detail on how the parental support measure was derived.

10. Methods: If possible, please include the PISA questions used to obtain the data on measures and covariates as a supporting information file.

11. Results: Line 258-259: Please clarify this sentence, as the footnote of Table 2 suggests that there are differences: “There were no statistically significant differences in the distribution of individual characteristics between boys and girls.”

12. Results: Line 260-261: It may be useful to provide a range here for prevalence of grade repetition.

13. Results: Line 290-292: Please provide P values in addition to 95% CIs for sensitivity analyses.

14. Discussion: Line 302-304: Please avoid the use of causal language, here and throughout the Discussion.

15. Discussion: Line 338: “The effect of grade repetition on bullying victimization is negative in our study.” Please avoid implying causality in this sentence.

16. Discussion: Line 381: Please remove the phrase “if causal” from this sentence, as your study cannot inform on the causal nature of this relationship.

17. Discussion: Line 383-384: Please revise to avoid causal implications: “Our study provides robust evidence for the negative consequence of grade repetition on bullying victimization.”

18. Discussion: Line 393-396: Please revise, interpreting the study based on the results presented in the abstract, emphasizing what is new without overstating your conclusions. “Our findings confirm that grade repetition is associated with bullying victimization and highlight that social promotion is a better policy practice than grade retention regarding to bullying victimization although both practices are inefficient 396 to improve educational outcomes of students.”

19. Please provide titles and legends for each individual table and figure in the Supporting Information.

20. Table 2 and Table 3: Please provide the statistical results (95% CIs and P values ) that are missing from the Table.

21. S3 Table: Please provide individual P values for each result. 

Comments from the reviewers:

Reviewer #1: This study aims to investigate the global association of grade repetition with bullying victimization in a large-scale school-based cross-sectional study.

Comments:

"All participants in Japan, Malaysia and Norway were excluded because no students in the three countries have ever repeated a grade (n=18,033). "

Can the authors please assess the bullying in these three countries for comparison of prevalence/frequency, given no grade repetition?

"Last, the participants without the data on grade repetition or school bullying (n=111,017) were excluded from the analysis"

Can the authors please discuss whether these excluded participants can be considered to be missing at random (MAR)?

The authors have followed STROBE and suitably included the associated checklist with the supplementary material.

"Participants were categorized as retained students if they answered "Yes" to any of these three questions, and promoted students if they didn't have the experience with repeating a grade at school before the survey."

Did the authors consider keeping "Yes, once" or "Yes, twice or more" as separate groups for the analysis, to explore the possibility of a dose-response relationship?

"The response options were "never or almost never", "a few times a year", "a few times a month", or "once a week or more". For each of the six experiences mentioned above, we identified bullying victimization by the responses of "a few times a month" or "once a week or more" [25, 26]. Besides, we combined the six experiences into a single indicator, "any type of bullying victimization", if the participants reported being bullied frequently (at least a few times a month) in any type".

Did the authors complete any sensitivity analyses looking at different ways to dichotomise bullying victimisation? Furthermore, did they consider keeping all four categorical outcomes within the analysis? 

"Demographic characteristics assessed included sex, age group, immigration status, school type, economic, social and culture status (ESCS), and perceived parents' emotional support".

Did the authors consider including BMI, disability, ethnicity, or urban/rural school location as other potential confounders in their analysis?

"As suggested by a study of PISA 2009 [2], we generated a dichotomous variable "age group" with 1 indicating the age is below the country average age, and 2 indicating the age is at or above the country average age".

Did the authors consider completing additional analyses treating age as a continuous variable?

"Finally, we adopted multivariable logistic regression models to estimate the strength, direction, and statistical significance of the relations linking experience with grade repetition to each type of bullying victimization by sex, adjusting the participating country ID and the measured control variables aforementioned in covariates assessment".

Did the authors consider applying hierarchical or multilevel modelling to better allow for clustering by country differences? Furthermore, there may be school differences within countries?

"The recently proposed E-value quantitates the minimum strength of association with both the exposure and the outcome which would be required of an unmeasured confounder to fully explain away the observed effect [30, 32]." 

This is a novel and useful approach for estimating the possible strength of impact of unknown confounding on the study inferences. Can the authors please clarify within the manuscript if this analysis considers one confounder at a time, or multiple confounders working together?

The authors are correct to state that the study design means that causality cannot be inferred. 

Therefore, the evidence provided from this study cannot determine whether grade repetition increases the chance of bullying, or bullying victimisation increases chance of grade repetition.

To introduce analyses that take into account the timing of events is a sensible idea, that might help uncover the direction of association that is observed in the study findings: "The reference period of being bullied (past 12 months before PIAS 2018) and repeating a grade at ISCED3 may overlap. To establish the temporal relationship between grade repetition and bullying victimization, we further repeated the analyses of the associations of repeating a grade at ISCED1 or ISCED2 with the risk of being bullied." 

However, this does not allow for occurrences of bullying prior to the last 12 months, which is not captured by the data.

The authors state in the results that: "As a sensitivity analysis we also evaluated associations of repeating a grade at ISCED1 or ISCED2 with the risk of being bullied to make sure exposure (grade repetition) preceded outcomes (bullying victimization), and found similar results (S3 Table). For instance, the multivariable adjusted OR (95% 291 CI) for bullying victimization in any type comparing promoted students to retained students in all students was 1.44(1.34-1.55); in girls: 1.51(1.34-1.70); and in boys: 1.40(1.29-1.52). " 

This is misleading as in fact this analysis does not 'make sure exposure (grade repetition) preceded outcomes (bullying victimization)'. Occurrences of bullying victimisation preceding ISCED1 or ISCED2 have not been captured in the data included within this analysis. Even more entwined, the relationship between grade repetition and bullying victimisation could be iterative in nature.

Furthermore, the conclusions that ""Globally the policy of grade repetition is harmful to students' health and wellbeing in some way regarding to school bullying" cannot and should not be inferred from the study outcomes found from this research. 

The authors acknowledge that: "The major limitation is this study only included students attended the school, which may be subject to possible selection bias. In addition, the cross-sectional design hinders us from establishing causality between grade repetition and bullying victimization", but later state that "however, we also used a finer category of repeating a grade before ISCED3 to ensure grade reputation preceded bullying victimization, and further examined the temporal direction of the relationship between grade repetition and bullying victimization", which is inaccurate as temporal direction is not ensured by the analysis due to the outcome question asking about the last 12 months only.

Similarly, the authors go on to state that: "Our findings, if causal and generalizable to broader age groups, may have significant implications for the wellbeing of 1.3 billion school-age children and adolescents around the world in an era embracing evidence-based interventions [47]. Our study provides robust evidence for the negative consequence of grade repetition on bullying victimization. The observed link with grade repetition and bullying victimization implies that the widespread educational policy of grade repetition may partly contribute to the differences in bullying victimization, and adds to the evidence against the policy of grade repetition."

This needs to be completely reworded, with the authors applying far more caution and care with their stated study implications and conclusions. Causality cannot be inferred by this analysis, and therefore causality should not be implied in the interpretation of the outcomes.

And again, "Our findings... highlight that social promotion is a better policy practice than grade retention regarding to bullying victimization although both practices are inefficient to improve educational outcomes of students."

This statement is not reflective of the evidence presented here.

Reviewer #2: I like the idea of looking at grade retention and bullying victimization using the PISA data. I have a few concerns to consider.

1. I would not look at bullying victimization by type. We have shown as have others that what matters for outcomes is severity and not form (see Haltigan & Vaillancourt, 2018). You also improve measurement precision by creating a composite score as suggested by Vaillancourt et al., 2010. 

2. I would concentrate my efforts on looking at grade retention by bullying victimization by country. You have this in your S2 table; it should replace your table on type. 

3. The implied direction is that being retained impacts bullying but it is possible that being bullied impact grades which impacts retention. There are longitudinal studies supporting both pathways bullied to poor grades and poor grades to being bullied. This needs to be addressed in the manuscript. 

4. Boys are more likely than girls to be retained than girls and to be bullied. I would look at this as a 3-way interaction. Maybe even a 4-way as per suggestion #2. 

5. There are a number of typos that need to be addressed (example lines 107-108). Avoid using contractions like isn't or didn't. 

Reviewer #3: The manuscript, Grade repetition and bullying victimization in teenagers: A global cross-sectional study, examines the association between a history of grade retention and experiences of bullying victimization in the past year for adolescents with an average age of about 15.8 years in 74 countries around the world. The study addresses a relatively neglected area of grade retention research by examining its association with bullying victimization. Strengths and weaknesses include the large, global sample. Strengths include the inclusion of so many countries around the world with a large sample size. Weaknesses include the averaging of the experiences of adolescents from such diverse countries with different cultures related to school experiences, peer relationships, and gender. The paper also would benefit from significant copy editing due to numerous typos, run-on sentences, and errors in word selection and grammar. 

The authors build their study rationale based on a lack of empirical studies relating grade retention and bullying victimization. They touch on some reasons why other researchers have explained associations or no associations between grade retention and bullying victimization. However, the study is not guided by a theoretical orientation or explanation of the mechanisms why grade retention would be related to bullying victimization. I recommend authors add an orientation and then also attend to how this orientation would relate to various cultural contexts around the globe. Would associations differ in different cultural contexts? Is an analysis by country warranted? Why or why not?

At the bottom of page 10, the authors wrote, "Finally, we adopted multivariable logistic regression models to estimate the strength, direction, and statistical significance of the relations linking experience with grade repetition to each type of bullying victimization by sex, adjusting the participating country ID and the measured control variables aforementioned in covariates assessment." I was unclear how the authors used the participating country ID in the analysis. I am also curious why authors did not report the full results of the logistic regression in terms of the association of control variables to bullying victimization. 

Authors state that their research goal is to understand the association between grade repetition and bullying victimization, but they also touch upon gender differences. It would be helpful for authors to move from a theoretical orientation to the status of empirical evidence for their purpose to a final, complete set of research questions and hypotheses. Authors examine overall bullying, single bullying experiences (measured and not latent constructs), and gender differences. They should explain also why a global average is best and why examining individual country contexts is not needed. 

In their discussion, authors wrote, "The effect of grade repetition on bullying victimization is negative in our study" but, in fact, grade repetition was positively related to bullying victimization (students who had been retained were more likely to experience bullying victimization). Also, authors should avoid using terms like "effect" that imply a causal association.

[LINK]

---

## [Decision Letter · Decision Letter 2]

19 Jul 2021

Dear Dr. Zuo,

Thank you very much for re-submitting your manuscript "Grade repetition and bullying victimization in teenagers: a global cross-sectional study" (PMEDICINE-D-21-00390R2) for consideration in PLOS Medicine’s Special Issue: Global Child Health: From Birth to Adolescence and Beyond. 

Your revised paper was evaluated by a senior editor and discussed among all the editors here. It was also discussed with the Special Issue Guest Editors, and evaluated by two of the original reviewers, including a statistical reviewer. The reviews are appended at the bottom of this email and any accompanying reviewer attachments can be seen via the link below:

[LINK]

Given the remaining requests of Reviewer 3, we would like to consider a further revised version that addresses the reviewers' and editors' comments. Obviously we cannot make any decision about publication until we have seen the revised manuscript and your response, and we may to seek re-review by one or more of the reviewers. 

We expect to receive your revised manuscript by Jul 30 2021 11:59PM. Please email us (plosmedicine@plos.org) if you have any questions or concerns.

We look forward to receiving your revised manuscript. 

Sincerely,

Caitlin Moyer, Ph.D.

Associate Editor 

PLOS Medicine

plosmedicine.org

1. Title: Please capitalize the first word of the subtitle: “Grade repetition and bullying victimization in teenagers: A global cross-sectional study”

2. Abstract: Methods and Findings: Please note the PISA assessment is administered to all students attending secondary education, with ages ranging from 15-16 years. Please clarify what is meant by “deprived victimization score” by briefly describing this is a composite score. 

3. Abstract: Methods and Findings: Line 38: Please note the potential confounders adjusted for in the analysis.

4. Abstract: Methods and findings: Line 48: Please briefly elaborate on the sex differences observed.

5. Abstract: Conclusions: Please clarify the final sentence, we suggest: “Sex differences in risk of experiencing bullying suggest that girls who repeat a grade may benefit from tailored interventions.” or similar.

6. Author summary: Please revise the authors summary. Please provide 2-3 single sentence bullet points under each of the following headings. Bullet points should be objective, brief, succinct, specific, accurate, and avoid technical language.

-Why Was This Study Done? Authors should reflect on what was known about the topic before the research was published and why the research was needed.

-What Did the Researchers Do and Find? Authors should briefly describe the study design that was used and the study’s major findings. Do include the headline numbers from the study, such as the sample size and key findings. 

-What Do These Findings Mean? Authors should reflect on the new knowledge generated by the research and the implications for practice, research, policy, or public health. Authors should also consider how the interpretation of the study’s findings may be affected by the study limitations.

7. In text references: Please include a space between the word and the reference bracket. For multiple references listed within brackets, please do not include a space (for example [13,15]).

8. Introduction: Line 76: Please change “Belgian” to “Belgium” and please revise to read “...while grade repetition has been discouraged strongly…”

9. Methods: Line 154-155: From looking at the Figure 1 flowchart, it is not clear whether in these countries grade repetition does not occur, or whether the data were not collected.

10. Methods: Line 188: Please provide more information on the construction of the composite victimization score. 

11. Methods: Line 216: Please define the abbreviation ESCS at the first point of use in the text.

12. Methods: Line 240: Please indicate any changes made to the analysis in response to peer reviewer comments (such as the implementation of the victimization composite score).

13. Methods: Is it possible to include a supporting information file describing the average ages by country?

14. Results: Line 289: Please provide the exact p values for the sex-specific analyses describing victimization score differences for students who were retained compared to those who were promoted.

15. Results: Line 314: Please change “regress” to “regression” in this sentence. Please quantify the results from this analysis with 95% CIs and p values.

16. Discussion: Line 364-365: We suggest revising to “...may be a contributing factor to the relationship between grade repetition and bullying…” or similar to avoid causal implications.

17. Discussion: Line 378: We suggest revising to “These findings suggest that girls may benefit from targeted attention in anti-bullying programs.” or similar.

18. Discussion: Line 384-385: Please clarify what is meant by “...because the actual cross-national comparable data on bullying can only be collected by pictures.”

19. Discussion: Line 400: Please replace “improve” with “increase” or similar.

20. Page 20: Please remove the Data Availability, Funding, Competing Interests sections from the main text, and ensure all information is accurately entered in the Data Availability, Financial Disclosure, and Competing Interests sections of the manuscript submission metadata.

21. References: Please use the "Vancouver" style for reference formatting, and see our website for other reference guidelines https://journals.plos.org/plosmedicine/s/submission-guidelines#loc-references

Please check the abbreviations for Journal titles: For example, reference 36 should be Lancet Child Adolesc Health.

22. Figure 1: Please clarify whether Japan, Malaysia and Norway did not measure/report data on grade repetition, or whether grade repetition does not occur in these countries (Methods: Line 154-155: “All participants in Japan, Malaysia, and Norway were excluded because no students in the three countries have ever repeated a grade (n=18,033).”

23. Table 2 and Table 3: Please do include the versions of the table, with 95% CIs and p values reported for each comparison for bullying victimization and retention between boys and girls.

24. Table 4: Please also include the results from the unadjusted models. Please provide the 95% CIs as upper and lower bounds, as well as the p values for each comparison.

25. S3 Table: Please remove all trademark/copyright symbols from the table (e.g. <iPad®>,<BlackBerry® PlayBookTM>)

26. S4 Table, S7 Table, and S10 Table: Please include the results of the unadjusted analyses as well.

Comments from the reviewers:

Reviewer #1: The authors have responded to each comment in turn, substantially rewording the manuscript where required.

Reviewer #3: The manuscript, Grade repetition and bullying victimization in teenagers: A global cross-sectional study, examines the association between a history of grade retention and experiences of bullying victimization in the past year for adolescents with an average age of about 15.8 years in 74 countries around the world. The study addresses a relatively neglected area of grade retention research by examining its association with bullying victimization. Overall, the paper is stronger but still has some areas that could be improved with further work including strengthening the theoretical rationale and clarifying the measurement and bullying victimization composites. The paper would still benefit from additional significant copy editing due to numerous typos and errors in word selection and grammar. 

Authors mention that several theoretical orientations have been proposed to explain why retained students are at higher risk of being bullied but doesn't really describe a theory - authors could go more in depth in detailing a theory that drives their selection of variables. What is missing for me is thought put into how cultural differences between countries may impact the association. For example, authors mention that the degree of stigma of repeating a grade varies by culture - can the author provide some key examples? The theory should be connected to the selection of variables and the interpretation of results. I want to know why authors feel there was an association in some but not all countries. Are there cultural or policy or prevalence differences? 

- Authors describe two previous studies - one in Brazil and one in the U.S. - how did the results differ from the Brazil and U.S. results in this study? 

- Are higher retention rates related to less association between retention and victimization?

- I wonder if the limitation of the sample only being students attending school differs by country as well. Are there differences in school attendance rates for 15year-olds around the globe? 

Regarding the measurement of victimization, I am a bit unclear on why all three types of measures are needed. Looking at each type and any type of victimization makes sense. How does the composite mean score add value? Would this be a "diversity" of bullying victimization experience? It seems hard to validate such a composite b/c someone could be physically attacked every day but never experienced any of the other types of victimization and they would have a lower score than someone who has been teased a few times a month and excluded a few times a month. The terms "victimization prevalence" and "victimization score" don't help with understanding the differences. I would appreciate a rationale provided for each measure along with a citation with validity evidence to support using this measure from extant literature. 

When describing the participants, it doesn't seem helpful to write that half the participants are lower than the average age (delete) since that is self-evident. 

In their interpretation of results, authors should discuss the possibility that bullying victimization caused the grade repetition, unless their theoretical orientation or existing empirical literature strongly support a one directional relation. 

There were too many types and wording errors to document but here are a couple of them: 

- What is "deprived victimization score"? Perhaps authors mean, "derived"

- In third author summary bullet point, I recommend changing "and vary" for "yet prevalence varies"

- I don't understand the sentence on page 18 lines 383-385 "Second, the cultural differences in the bullying definition also partly responsible for the variation because the actual cross-national comparable data on bullying can only be collected by pictures."

Additional minor wording recommendations: 

- Delete, "as we mentioned in the background section."

[LINK]

---

## [Decision Letter · Decision Letter 3]

26 Aug 2021

Dear Dr. Zuo,

Thank you very much for submitting your manuscript "Grade repetition and bullying victimization in teenagers: A global cross-sectional study" (PMEDICINE-D-21-00390R3) for consideration in PLOS Medicine’s Special Issue: Global Child Health: From Birth to Adolescence and Beyond. 

Your revised paper was evaluated by a senior editor and discussed among all the editors here. It was also discussed with an academic editor with relevant expertise, and sent to one of the original reviewers. The reviews are appended at the bottom of this email and any accompanying reviewer attachments can be seen via the link below:

[LINK]

In light of the remaining comments from the reviewer and academic editor, we will not be able to accept the manuscript for publication in the journal in its current form, but we would like to consider a further revised version that addresses the reviewers' and editors' comments. Obviously we cannot make any decision about publication until we have seen the revised manuscript and your response, and we plan to seek re-review by one or more of the reviewers. 

We expect to receive your revised manuscript by Sep 02 2021 11:59PM. Please email us (plosmedicine@plos.org) if you have any questions or concerns.

We look forward to receiving your revised manuscript. 

Sincerely,

Caitlin Moyer, Ph.D.

Associate Editor 

PLOS Medicine

plosmedicine.org

1. From the Academic Editor: Please cite the source of the bullying composite score, indicate whether or not the score has been validated in other study populations, the alpha level of reliability of the composite score in this study population to ensure the composite score is reflective of a single, underlying bullying construct. This information is necessary to support that the scale is sufficiently reliable to be used as a single composite.

2. Abstract: Methods and Findings Line 26-27: Please revise to: “...the limited evidence of associations between grade repetition on school bullying is inconsistent…”

3. Abstract: Line 51: Please revise to “...having possessions taken away…” or similar.

4. Abstract: Conclusions: We suggest opening the first sentence with “In this study, we observed that globally, both boys and girls who ever repeat…”

5. Author summary: Please format these as bullet points.

6. Author summary: Line 79: Please revise to “Grade repetition was associated with bullying victimization in both boys and girls.”

7. Author summary: Line 87: Please revise to: “Girls who repeat a grade may be at higher risk of experiencing bullying than boys, and may benefit from targeted interventions.” or similar.

8. Introduction: Line 105: Please reference this statement: “...a common misperception is that repeating a grade allows the student to grow academically and socially.”

9. Introduction: Line 144: This is the first instance where sex differences are mentioned. It would be helpful to have some rationale for your hypothesis that the association between bullying and repetition would be stronger among girls.

10. Methods: Line 166: Please cite the PISA website as a reference.

11. Methods: Please provide more detail here or in the supporting information files (such as S3 Table) describing how the ESCS and parental emotional support scores were calculated from the survey responses.

12. Methods: Line 210: Please address the comments of the Academic Editor and Reviewer 3. We require that you please provide necessary information to support the use of the composite victimization score, including information pertaining to validity and rationale for its use, and please discuss the limitations of the score.

13. Results: Line 310- : Please provide the p values in addition to the 95% CIs for the adjusted ORs for the associations of each type of bullying with school promotion.

14. Results: Line 333-334: Please clarify this sentence: “Across all countries, retained students reported a higher level and higher prevalence of victimization than promoted students.” or similar, depending on your meaning.

15. Conclusions: Line 452-453: Please avoid the causal language in the final sentence, and please revise to: “Grade repetition was associated with increased likelihood of bullying victimization…”

16. References: Please check that the citation for reference 1 is complete.

17. S8 Table: Please remove the extra parentheses at the top of the “Adjusted” column under Odds ratio.

Comments from the reviewers:

Reviewer #3: Overall the paper is stronger in this revision, however, the authors did not adequately address my concerns about the measurement of victimization. The construct of bullying cannot be measured by creating a sum score of frequency responses to a diversity of types of victimization experiences and using it as a continuous item. I do not believe a reviewer was justified in recommending this practice, it does not make conceptual sense as bullying victimization experiences (e.g., verbal, physical, relational, cyber) do not all need to occur for someone to be severely bullied. The authors cite Vaillancourt et al. (2010) as recommending this process but, from my reading, their research recommended using a measure of behavioral based bullying with a cut-off score for a binary determination of yes/no bullied with no=never and yes=any bullying. Thus, if the authors and editor feel that the reviewer feedback is important to comply with and that a composite measure of bullying victimization is critical to answering their research questions, they should provide adequate explanation for and documentation of why this approach is valid and what this approach is measuring. I recommend Green et al. (2013) and a couple of additional articles about measuring bullying in informing the authors' decision. 

Green, J. G., Felix, E. D., Sharkey, J. D., Furlong, M. J., & Kras, J. E. (2013). Identifying bully victims: Definitional versus behavioral approaches. Psychological Assessment, 25, 651-657.

Sharkey, J. D., Ruderman, M. A., Mayworm, A. M., Green, J. G., Furlong, M. J., Rivera, N., & Purisch, L. (2015). Psychosocial functioning of bullied youth who adopt versus deny the bullied victim label. School Psychology Quarterly, 30, 91-104. 

Felix, E. D., Sharkey, J. D., Green, J. G., Furlong, M., F., & Tanigawa, D. (2011). Getting precise and pragmatic about the assessment of bullying: The development of the California Bullying Victimization Scale. Aggressive Behavior, 37, 234-247. 

The rest of my feedback was adequately addressed.

[LINK]

---

## [Decision Letter · Decision Letter 4]

20 Sep 2021

Dear Dr. Zuo,

Thank you very much for submitting your manuscript "Grade repetition and bullying victimization in teenagers: A global cross-sectional study" (PMEDICINE-D-21-00390R4) for consideration in PLOS Medicine’s Special Issue: Global Child Health: From Birth to Adolescence and Beyond. 

Your paper was evaluated by a senior editor and discussed among all the editors here. It was also discussed with an academic editor with relevant expertise, and re-evaluated by one of the reviewers. The reviews are appended at the bottom of this email and any accompanying reviewer attachments can be seen via the link below:

[LINK]

In light of the comments from the academic editor, we will not be able to accept the manuscript for publication in the journal in its current form, but we would like to consider a revised version that addresses the editors' comments. Obviously we cannot make any decision about publication until we have seen the revised manuscript and your response. 

We expect to receive your revised manuscript by Sep 27 2021 11:59PM. Please email us (plosmedicine@plos.org) if you have any questions or concerns.

We look forward to receiving your revised manuscript. 

Sincerely,

Caitlin Moyer, Ph.D.

Associate Editor 

PLOS Medicine

plosmedicine.org

1. From the academic editor: Given that the authors report an alpha reliability for the scale of 0.88, it is unclear why they removed the composite scale. It also is unclear whether the alpha that is reported is from the original article describing the scale or whether it is from the sample under study. If the former, this should be stated, and the alpha for the sample should be provided. If the latter, it is unclear why the authors do not use a composite score, as an alpha of 0.88 is adequate evidence justifying the use of a composite score. In sum, the descriptions of the outcome measure needs to be cleaned up and the justification for using whatever measurement scale for the outcome(s) needs to be more clearly justified.

2. Title: We suggest revising the title to “Grade repetition and bullying victimization in adolescents: A global cross-sectional study of the Program for International Student Assessment (PISA) data from 2018.

3. Throughout: If possible, please replace “retained students” or “retained girls” etc. with “students who repeat a grade” or similar.

4. Abstract: Lines 24-25: Please clarify what is meant by primary in the sentence “Globally around 32.2 million students repeated a grade in primary in 2010.” Please revise if this should read “in primary school” or “in the primary education level” or similar.

5. Abstract: Line 49: Please revise “were associated with” to “retained girls had higher risks of” or similar.

6. Abstract: Line 58: Please revise “from repeating a grade than boys” to “...risks of specific types of bullying associated with repeating a grade than boys.”

7. Author summary: Line 84: We suggest revising to “The experience of repeating a grade may suggest a need for bullying interventions among both boys and girls”

8. Author summary: Line 86-87: We suggest revising to “Girls who repeat a grade may be at higher risk of experiencing some forms of bullying than boys, and may benefit from targeted interventions.”

9. Methods: Lines 198-209: As noted by the academic editor, please clarify whether the alpha here represents the original study, and provide the alpha for the sample if so. Please provide a clear explanation and rationale for the bullying victimization outcome used.

10. Methods: Lines 215-218, 245-246; 253-254: Reference 33 cited here is listed as “forthcoming” in the reference list. There are a number of details of the methods that reference this technical report, please update the reference in the event the 2018 report is available now, or otherwise clarify how the PISA 2018 technical report may be accessed.

11. Results Lines 314-317: “The sex-specific analyses produced similar results in boys and girls (Table 4). Furthermore, interaction analyses showed that retained girls were at higher risk of being made fun of, being threatened, having possessions taken away, and being pushed around compared with retained boys (S5 Table).” We suggest making it more clear that while sex differences were not apparent overall for bullying victimization, there were interactions indicating that for subtypes of bullying, there were differences between girls and boys who repeated a grade.

12. Discussion: Line 376: Please remove the word “undoubtedly” from the sentence, and please revise “As predicted” to “Consistent with previous work” or “Consistent with our hypothesis” or similar.

13. Discussion: Line 394: Please revise “than the promoted low achievers” to “than students who are promoted but who may have lower academic achievement” or similar.

14. Conclusion: Line 444: Please revise to avoid causal implications, we suggest: “girls are more likely than boys to experience specific types of bullying associated with repeating a grade.”

Comments from the reviewers:

Reviewer #3: I appreciate your careful consideration of the composite of bullying and ultimate removal of the composite from the study.

[LINK]

---

## [Editor Report · Decision Letter 5]

6 Oct 2021

Dear Dr. Zuo,

Thank you very much for re-submitting your manuscript "Grade repetition and bullying victimization in adolescents: A global cross-sectional study of the Program for International Student Assessment (PISA) data from 2018" (PMEDICINE-D-21-00390R5) for consideration in PLOS Medicine’s Special Issue: Global Child Health: From Birth to Adolescence and Beyond.

I have discussed the paper with my colleagues and the academic editor. I am pleased to say that provided the remaining editorial and production issues are dealt with we are planning to accept the paper for publication in the journal.

The remaining issues that need to be addressed are listed at the end of this email. Please take these into account before resubmitting your manuscript:

[LINK]

In revising the manuscript for further consideration here, please ensure you address the specific points made by the editors. In your rebuttal letter you should indicate your response to the editors' comments and the changes you have made in the manuscript. Please submit a clean version of the paper as the main article file. A version with changes marked must also be uploaded as a marked up manuscript file.

We look forward to receiving the revised manuscript by Oct 13 2021 11:59PM.   

Sincerely,

Caitlin Moyer, Ph.D.

Associate Editor 

PLOS Medicine

plosmedicine.org

Requests from Editors:

1. Abstract: Conclusions: Line 59-61: Please remove the word “robust” at line 59. We suggest revising the final sentence slightly to: “Sex differences in risk of experiencing some subtypes of bullying suggest that tailored interventions for girls who repeat a grade may be warranted.” or similar, as the study findings do not seem to exactly imply that girls could benefit more than boys from a future intervention.

2. Methods: Line 207-211: Please clarify the description slightly that for the six individual bullying experiences the response options for 0= never or almost never and 1=a few times a year were combined into 0 (a few times a year or less frequently) and the response options for 2= a few times a month and 3= once a week or more were combined into 1 (a few times a month or more frequently). For the combined measure, please make it clear if “0 (never on all types)” corresponds to the “never or almost never” original response to the question, and 1 (any involvement) corresponds to the combination of responses 1, 2, or 3.

3. Methods: Line 225-229: We suggest replacing “immigrant” with “migrant” as this term may imply a more neutral point of view. Please replace immigrant/immigration status throughout, including Table 1.

4. Results: Line 291: We suggest avoiding the term “immigrant” and revising to “...reported a non-native status” or similar.

5. Discussion: Line 344-345: We suggest revising slightly to: “Third, girls who repeated a

grade are more likely to report having experienced specific types of bullying than boys.”

6. Discussion: Line 363: Please revise this sentence if this should read “...have so far focused on its academic outcomes…”

7. Discussion: Line 400-401: We suggest revising this sentence slightly: “The higher risk of experiencing some types of bullying among girls who repeated a grade suggests the possibility that targeted anti-bullying prevention interventions may be especially beneficial for girls.” or similar.

8. Discussion: Lines 438-441: We suggest revising the sentence to read: “The observed link raises the possibility that the widespread educational policy of grade repetition may partly contribute to the differences in bullying victimization and adds to the evidence against the policy of grade repetition; however, our study cannot establish a causal relationship.” or similar.

9. Social Media: To help us extend the reach of your research in the event that your article is accepted for publication, please provide any Twitter handle(s) that would be appropriate to tag, including your own, your coauthors’, your institution, funder, or lab.

[LINK]

---

## [Editor Report · Decision Letter 6]

11 Oct 2021

Dear Dr Zuo, 

On behalf of my colleagues and the Academic Editor, Kathryn Yount, I am pleased to inform you that we have agreed to publish your manuscript "Grade repetition and bullying victimization in adolescents: A global cross-sectional study of the Program for International Student Assessment (PISA) data from 2018" (PMEDICINE-D-21-00390R6) in PLOS Medicine’s Special Issue: Global Child Health: From Birth to Adolescence and Beyond.

PRESS

Sincerely, 

Caitlin Moyer, Ph.D. 

Associate Editor 

PLOS Medicine